# Observation of E-cadherin adherens junction dynamics with metal-induced energy transfer imaging and spectroscopy
Tao Chen [1] ✉, Narain Karedla [2,3] & Jörg Enderlein [1,4] ✉

Epithelial cadherin (E-cad) mediated cell-cell junctions play a crucial role in the establishment and maintenance of tissues and organs. In this study, we employed metal-induced energy transfer imaging and spectroscopy to investigate variations in intermembrane distance during adhesion between two model membranes adorned with E-cad. By correlating the measured intermembrane distances with the distinct E-cad junction states, we probed the dynamic behavior and diversity of E-cad junctions across different binding pathways. Our observations led to the identification of a transient intermediate state referred to as the X-dimeric state and enabled a detailed analysis of its kinetics. We discovered that the formation of the X-dimer leads to significant membrane displacement, subsequently impacting the formation of other X-dimers. These direct experimental insights into the subtle dynamics of E-cad-modified membranes and the resultant changes in intermembrane distance provide perspectives on the assembly of E-cad junctions between cells. This knowledge enhances our comprehension of tissue and organ development and may serve as a foundation for the development of innovative therapeutic strategies for diseases linked to cell-cell adhesion abnormalities.

Epithelial cadherin (E-cad) is a prominent member of the classical cadherin family, categorized as a type I cadherin. It serves a crucial role in facilitating calcium-dependent adhesion among epithelial cells, thereby contributing to the formation of adherens junctions on the cell surface[1–4]. By mediating the adhesion between epithelial cells, E-cad contributes to the establishment and maintenance of cellular interactions within tissues. The formation of adherens junctions, enabled by E-cad's activity, promotes the cohesion and structural stability necessary for the proper functioning of organs and tissues.

However, the regulation of E-cad expression and activity is not static but rather subject to dynamic changes. These alterations play a vital role in various physiological and pathological processes[5]. For instance, the loss or downregulation of E-cad expression is often observed in metastatic cancers, where it contributes to the acquisition of invasive and migratory properties by cancer cells[6]. In such cases, E-cad's normal adhesive function is disrupted, allowing cancer cells to detach from their original location and invade surrounding tissues.

Additionally, during inflammatory responses, the dysregulation of adherens junctions and E-cad expression can compromise the integrity of epithelial barriers. This disruption can result in increased permeability and allow the infiltration of harmful substances, exacerbating the inflammatory process. Thus, the dynamic regulation of E-cad in these scenarios becomes crucial for maintaining tissue homeostasis and preventing the progression of inflammatory disorders[7–11].

The classic form of E-cad is comprised of five structurally similar extracellular cadherin-like domains (EC domains) arranged in tandem. These domains are connected by conserved calcium-binding linker regions. E-cad also possesses a single transmembrane segment and a highly conserved cytoplasmic tail. Numerous studies have underscored the pivotal role of the EC domains of cadherin in regulating the assembly of cell-cell junctions[12–15].

The process of cell-cell adhesion involves multiple steps, encompassing the dynamic transition between various molecular states, ranging from free monomers to trans-dimers, clusters of E-cad molecules, and interactions with intracellular domains (Fig. 1a)[16]. A weak and transient X-dimer has been reported to exist during the adhesion process, acting as a kinetic intermediate that lowers the energy barrier during the formation of strand-swapped S-dimers in adherens junctions[12]. Initially, this X-dimer was identified through site-directed mutations in E-cad[17,18]. More recently, the X-dimer was finally observed in wild-type cadherins using cryo-electron microscopy[19].

[1]Third Institute of Physics—Biophysics, Georg August University, Göttingen, Germany. [2]The Rosalind Franklin Institute, Didcot, UK. [3]Kennedy Institute of Rheumatology, University of Oxford, Oxford, UK. [4]Cluster of Excellence 'Multiscale Bioimaging: from Molecular Machines to Networks of Excitable Cells' (MBExC), Universitätsmedizin Göttingen, Göttingen, Germany. ✉e-mail: tao.chen@phys.uni-goettingen.de; jenderl@gwdg.de

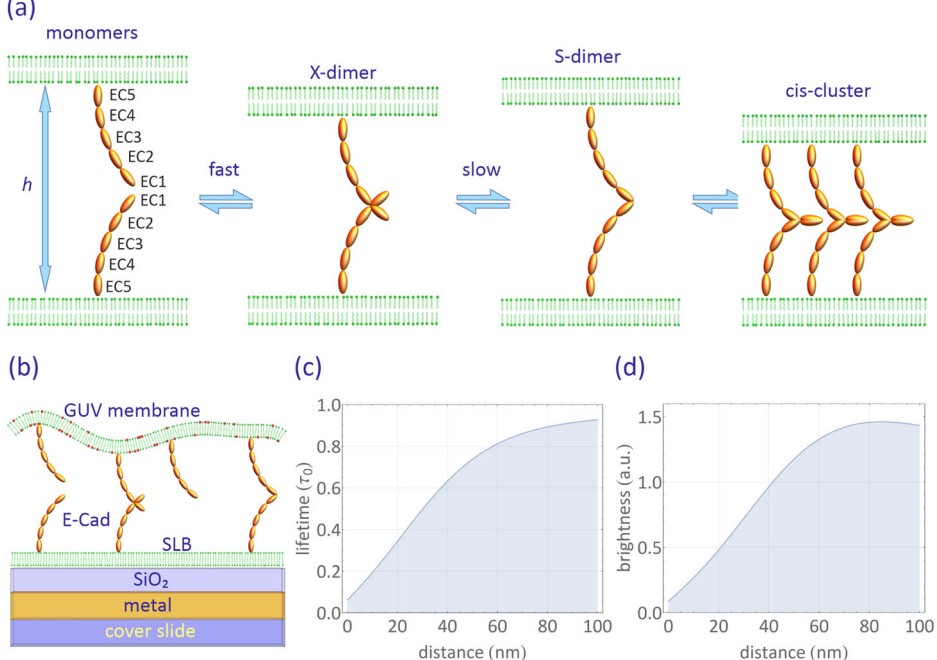

**Fig. 1 | Ecads model and MIET measurement. a** Schematic representation of classic E-cad states during clustering: This diagram illustrates the various states classic E-cads can assume during clustering. The process of trans-dimerization, where E-cads join together, is mediated by the extracellular subunits EC1 and EC2. It involves at least two distinct structural states: initially, a short-lived X-dimer forms from individual monomers, which then transition into a more stable S-dimer. The S-dimer has the ability to cluster further by establishing lateral interactions within the same cell, known as *cis*-interactions, mediated by EC1-EC2 connections to the actin cytoskeleton. These *cis*-interactions between E-cads on the same cell form a lattice-like structure as proposed in adherens junctions. Importantly, these transitions between molecular states can also occur in reverse, involving the disassembly of *cis*-interactions and transitions from S-dimers back to X-dimers and monomers.

**b** Experimental Configuration: This schematic illustrates the setup used in our experiment. E-cad proteins are positioned between a supported lipid bilayer (SLB) on a gold film, separated by a silica spacer, and a giant unilamellar vesicle (GUV) that is highly labeled with fluorescent dye. Calculated fluorescence (**c**) lifetime and (**d**) intensity dependence on distance: This graph displays the calculated relationship between fluorescence lifetime (expressed as $\tau/\tau_0$, with $\tau_0$ representing lifetime in free space) and fluorescence intensity relative to the distance above the substrate surface. The curves are calculated for a dipole with an emission wavelength of 690 nm and a random orientation. The substrate used for these calculations consists of multiple layers, including 10 nm silica, 1 nm titanium, 10 nm gold, 2 nm titanium, and a glass coverslip, arranged from top to bottom.

E-cads have been the subject of extensive investigation utilizing various experimental and theoretical approaches, which have significantly contributed to our understanding of E-cad-mediated intermembrane adhesion[12–14,16,20–26]. For example, NMR relaxation dispersion spectroscopy has enabled the measurement of kinetic parameters in the trans-dimerization of E-cad EC fragments in solution[12]. Single-molecule Förster resonance energy transfer studies have provided evidence of the mutual cooperativity of *cis/trans* interactions[13,21]. Fluorescence microscopy has revealed the involvement of a nucleation process in junction formation[23,27], while reflection interference contrast microscopy and computational simulations have shed light on the role of membrane fluctuations in mediating the interaction between E-cad bonds[22].

However, limited attention has been given to variations of membrane-membrane distance during adhesion. Crystal lattice studies have identified EC5-EC5 distances of 29 nm for the X-dimer, 37 nm for the S-dimer, and 19 nm for the *cis*-cluster of E-cad[19,28,29]. These variations in EC5-EC5 distances during adhesion would lead to a corresponding change in membrane-membrane distance, offering an alternative means of monitoring E-cad adhesion; however, resolving such minute distance changes requires a high-resolution technique.

Metal-induced energy transfer (MIET) imaging and spectroscopy represents a cutting-edge super-resolution fluorescence technique ideally suited for such inquiries[30–32]. MIET allows for localizing fluorophores in the axial dimension with nanometer-scale accuracy, employing conventional fluorescence lifetime imaging microscopy (FLIM). This technique capitalizes on the electrodynamic near-field coupling between an excited fluorophore and surface plasmons within a planar metal film. The distance-dependent energy transfer from the fluorophore (donor) to the metal (acceptor) in MIET leads to a distinctive modulation of both fluorescence lifetime and brightness as a function of the fluorophore's proximity to the metal surface. By measuring the fluorescence lifetimes and employing a suitable theoretical model, we can accurately determine the distance between the fluorophore and the metal.

We used MIET to dissect the intermembrane adhesion process mediated by E-cads. Our focus lies in exploring the variations in membrane height, particularly the membrane-membrane distance, during the adhesion of two E-cad-modified membranes. By employing this approach, we can unravel the dynamic landscape of E-cad binding states and provide information regarding the behavior and interactions of E-cads during cell-cell adhesion.

## Results

We used a biomimetic system to mimic Ecad-mediated cell adhesion, observing adhesion events between partially fluid supported lipid bilayers (SLBs) and giant unilamellar vesicles (GUVs). The SLBs were prepared using 1,2-dipalmitoyl-sn-glycero-3-phosphocholine (DPPC) doped with 1 mole% acyl chain-labeled 1-acyl-2-[12-[(7-nitro-2-1,3-benzoxadiazol-4-yl)amino]dodecanoyl]-sn-glycero-3-phosphocholine (NBD-PC), 5 mole% 1,2-dioleoyl-sn-glycero-3-[(N-(5-amino-1carboxypentyl)iminodiacetic acid)succinyl] nickel salt (Ni-NTA-DOGS), and 1,2-dioleoyl-sn-glycero-3-phosphoethanolamine-N-[methoxy(polyethylene glycol)-2000] (DOPE-PEG2000) by a Langmuir-Blodgett Langmuir-Schäfer (LB-LS) method. Note that the bottom leaflet of the SLB comprises only DPPC to ensure the flatness of the SLB on the MIET substrate. The GUVs were prepared using

**Fig. 2 | Two different adhesion scenarios. a** FLIM image of fluorescently-labeled GUVs containing E-cad on SLBs. The image was measured 1 h after incubation of the GUVs within the chamber. **b, c** Reconstructed height images for GUVs in different adhesion states. Height traces showing height variations during E-cad clustering for measurements on a partially fluid SLB (**d**), and for a fluid SLB (**e**). Inset at top-right in (**d**): Zoom-in of the trace marked by the blue rectangle.

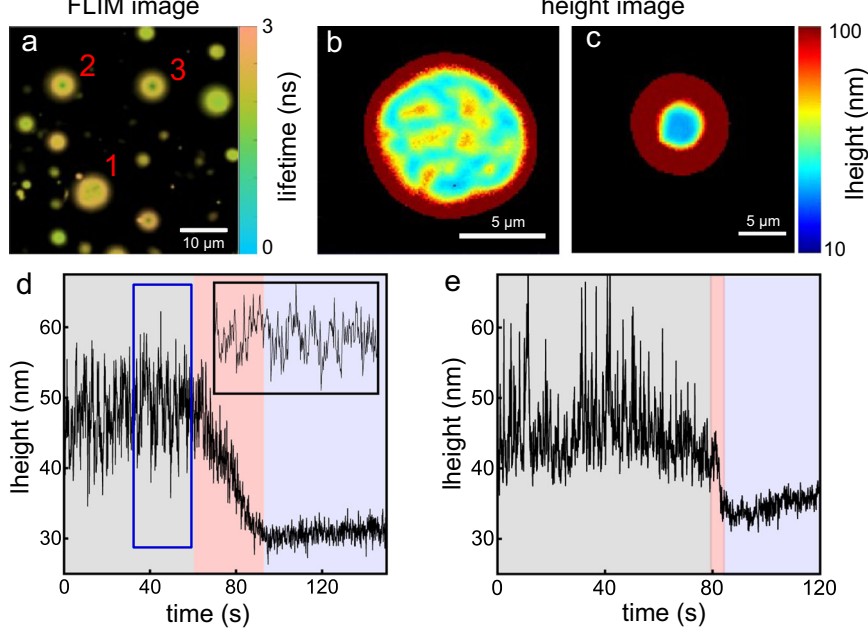

1-stearoyl-2-oleoyl-sn-glycero-3-phosphocholine doped with 5 mol% NTA-DOGS, 1 mol% DOPE-PEG(2000), and 1 mol% Atto655 headgroup-labeled 1,2-dipalmitoyl-sn-glycero-3-phosphoethanolamine (Atto655-DPPE). These membranes were decorated with E-cad extracellular (EC) domains (Fig. 1b). Control measurements confirmed the homogeneity of the SLBs and showed no evidence of phase separation in GUVs containing fluorescent lipid and Ni-NTA-DOGS molecules (Supplementary Fig. 1 and Supplementary Fig. 12). It should be noted that while the biomimetic system lacks the complexity of biological membranes and does not fully replicate the intricate behavior seen in cellular contexts, it enables us to observe E-cad interactions without the confounding influences of other cellular proteins, contextual factors, or active processes.

E-cad receptors were in the form of a dimeric chimera, with all five E-cad EC domains genetically fused to an immunoglobulin Fc-fragment exhibiting a hexahistidine tag (his6). The SLBs were prepared on a substrate consisting of a 10 nm gold film and a 10 nm silica spacer (denoted as MIET substrate). The distal leaflet of the SLB was doped with the Ni-nitrilotriacetic acid (NTA) chelating lipid, which can further bind to the his6 tags of the E-cad. The GUVs were also functionalized with E-cad EC at a concentration of 5 mol%. To induce membrane fluctuations and increase the excess area in the vesicles, we created an osmolarity difference between the inside of the GUVs (230 mOsm/L, sucrose solution) and the outer buffer (300 mOsm/L, phosphate-buffered saline, PBS, mixed with 0.75 mM $CaCl_2$ solution). No adhesion was observed in the absence of an osmolarity difference (Supplementary Fig. 3).

A high dye concentration (1 mol%, Atto655-DPPE) was used to eliminate the contributions form lateral diffusion to the observed fluorescence fluctuations: At such a high concentration, fluorescence intensity fluctuations arising from dye diffusion in and out of the confocal volume are negligible (Supplementary Fig. 2 and Supplementary Note 2)[33]. Furthermore, it's worth noting that no photobleaching was observed, attributed to the utilization of low excitation power for samples with high dye concentrations. The adhesion process was monitored using MIET imaging/spectroscopy. Figure 1c shows the calculated dependence of lifetime and brightness as a function of height from the silica surface (see section "Methods"). This allowed us to translate a measured lifetime into an axial distance value using this lifetime-distance calibration curve. In this study, only the membrane of the GUV was labeled, and its height over the surface was determined from the lifetime values of the labeling fluorophores.

The fluorescence imaging and lifetime measurements for the samples were conducted using confocal scanning microscopy, employing pulsed laser excitation and time-correlated single-photon counting (TCSPC) for fluorescence lifetime measurements, as detailed in the Materials and Methods section. For calculating FLIM, we constructed TCSPC curves for each pixel and fitted these curves with a multi-exponential decay model, giving us a mean fluorescence lifetime for each pixel[34]. Initially, we scanned the samples to identify suitable GUVs, including only those with a diameter exceeding 15 µm for further analysis.

In Fig. 2a, one can observe a fluorescence FLIM image of GUVs containing E-cad on the SLBs after 1 h of incubation. The FLIM image reveals two distinct binding states in the proximal membranes of the GUVs:

- The lifetimes and contact areas of the proximal membranes for GUV-2 and GUV-3 are significantly shorter and smaller than those of GUV-1.
- The lifetime distribution for the contact area is uniform for GUV-2 and GUV-3, while it displays more heterogeneity for GUV-1.

Utilizing the MIET calibration curve depicted in Fig. 1c, we transformed the FLIM images into height images. These height images revealed that the average bottom height of GUV-2 and GUV-3 above the surface is 35 ± 1 nm, while GUV-1 exhibits a greater height of 56 ± 9 nm. It's important to note that the fluorescent lipid molecules are distributed in both leaflets of the GUV. Therefore, the calculated height represents the distance between the center of the lipid bilayer of the GUV and the surface of the MIET substrate. Factoring in the thickness of the SLBs and GUVs (approximately 4 nm) as well as the hydration layer (about 2 nm)[31], we calculated that the membrane-membrane distance for GUV-2 and GUV-3 is approximately 27 nm, whereas for GUV-1, it measures roughly 48 nm. We assumed that GUV-2 and GUV-3 formed the *cis*-cluster junctions and calculated the EC5-EC5 distance by taking the size $d_{Fc}$ of the Fc fragment into account ($d_{Fc}$ = 7 nm, as determined from the crystal structure)[35], as well as the tilt angle of the E-cad in the clustered state (~30°, determined from the crystal structure)[25,28]. The estimated EC5-EC5 distance for GUV-2 and GUV-3 is around 20 nm, which closely aligns with the crystal structure of *cis*-cluster (19 nm), suggesting that GUV-2 and GUV-3 have indeed formed the *cis*-cluster junctions.

However, we could not precisely calculate the EC5-EC5 distance for GUV-1 because, unlike *cis*-cluster junctions, which have a well-defined orientation on the membrane[25,28], Ecad monomers (or Fc fragments) can be oriented at a range of angles on the membranes (or even lying flat on the membranes). Due to the observed inhomogeneity in GUV-1, classifying the

https://doi.org/10.1038/s42003-024-07281-4                                                   **Article**

binding state for GUV-1 solely based on its average height proves challenging. This observation implies that, in addition to the E-cads on the surface of the membranes (both GUV and SLB) that culminate in the final *cis*-clustered state (as seen in GUV-2 and GUV-3), some E-cads do not contribute to the formation of the ultimate *cis*-cluster. In fact, during our 2-h observation period, membranes such as GUV-1 consistently maintained this heterogeneous pattern. This observation aligns with a previous report indicating the existence of a nucleation process during E-cad junction formation, resulting in an all-or-nothing characteristic[23].

In the initial phase of *cis*-cluster interface formation, we continuously monitored the E-cad-mediated adhesion process via FLIM imaging, see Supplementary Movie 1. Initially, the giant unilamellar vesicles (GUVs) positioned themselves above the SLBs to establish a contact zone. This contact zone subsequently contracted during the adhesion process due to the the enrichment of *cis*-clusters. To track the height variations throughout this process, we performed point measurements to record fluorescence intensity traces (Supplementary Fig. 5).

Subsequently, we transformed these fluorescence intensity traces into height traces by binning them at a 100 ms resolution. For each time bin (containing between $1 \times 10^4$ and $5 \times 10^4$ photons), we constructed a fluorescence decay histogram and determined the fluorescence lifetime using a multi-exponential decay model. The obtained lifetimes were then converted into height values utilizing the calibration curve presented in Fig. 1c.

A typical height trace representing one GUV adhesion event is illustrated in Fig. 2d, which can be segmented into three distinct phases. The initial black region in Fig. 2d signifies the waiting period preceding E-cad clustering. The subsequent red segment corresponds to the clustering process itself, and the subsequent blue area represents the time trace of E-cad after clustering. It should be noted that the waiting times depicted in Fig. 2d, e do not precisely correspond to the initiation of adhesion. This discrepancy arises from the scanning of the sample before conducting the point measurement, leading to a delay in capturing the exact onset of adhesion.

During the course of the curve depicted in Fig. 2d, the membrane initially exhibited significant height fluctuations within the range of 45–55 nm (black region). Subsequently, it gradually decreased to a lower height of approximately 31 nm with reduced fluctuations (blue region). The initial height range of 45–55 nm corresponds to a minimum EC5-EC5 distance of 23–33 nm, assuming no tilt of the Fc fragment. In contrast, the final height of approximately 31 nm corresponds to an EC5-EC5 distance of 16 nm, assuming a tilt angle of 30 degrees for the E-cad at *cis*-cluster state. We attribute this to the conversion of Ecad from its dimeric state (S-X-dimer) to a final *cis*-clustered state. It is worth noting that we cannot directly observe monomer to dimer formation due to the fast dimerization rate $(3.8 \times 10^4\ s^{-1})$[16,20] of E-cad monomers. However, this does not preclude the existence of monomers in the initial state because E-cads dynamically transition between different states.

In Fig. 2d, the red segment highlights the phase of *cis*-clustering. We discovered that this clustering strongly depends on the mobility of the membrane. For instance, when we replaced the partially fluid SLB (composed of DPPC + NBD-PC) with a fluid SLB made of pure SOPC lipid (Supplementary Fig. 4 and Supplementary Note 3), the clustering rate increased by at least an order of magnitude (Fig. 2e, further detailed in Supplementary Fig. 5 and Supplementary Note 4).

In instances where cluster formation did not occur, the lower membrane of a GUV exhibited continuous fluctuations at a consistent height, as visually demonstrated in Supplementary Movie 2. In our MIET measurements, the GUV membrane fluctuates above the gold-coated surface, causing variations in fluorescence intensity (Fig. 1c). Importantly, we verified that these intensity fluctuations were the result of vertical displacements of the membrane and not caused by the lateral diffusion of individual fluorophores. This is due to the high fluorophore label concentration (1 mol %), which excluded any contribution from the lateral diffusion of the fluorophores to the observed intensity fluctuations (Supplementary Fig. 2 and Supplementary Note 2).

Furthermore, we conducted point measurements on this fluctuating membrane and analyzed the time trace of fluorescence intensity, which was constructed over time bins with a width of 10 ms. Surprisingly, we discovered that the intensity distribution exhibited an asymmetry, and numerous "dip" signals were evident in the intensity-time traces (Fig. 3a, and Supplementary Fig. 6). As a control, we did not observe such "dip" signals in a GUV membrane lacking E-cad or in a membrane after the formation of the *cis*-cluster (Fig. 3b).

Returning to the first case that led to the formation of the final *cis*-cluster, we also observed these "dip" signals in certain time traces within the waiting time interval preceding E-cad clustering (Fig. 2e, inset).

To unravel the origin of the "dip" signals, we employed a two-threshold method to demarcate the "dip signal" (referred to as the low state, blue area) from the baseline (referred to as the high state, red area) within the intensity-time trace (see Supplementary Fig. 8 and Supplementary Note 6). Subsequently, by accumulating decay times based on their photon counts for each time bin in the time trace, we constructed fluorescence lifetime decay histograms for these two distinct states and determined their respective fluorescence lifetimes (Fig. 3c).

By fitting the fluorescence decay curves using a multi-exponential model, we extracted lifetimes of 1.65 ns for the high state and 1.51 ns for the low state, respectively. These lifetimes corresponded to heights of 55 nm and 47 nm above the silica surface. We analyzed 45 fluorescence time traces, each lasting for 5 min, from 27 GUVs. For the heights of the high state and low state, we obtained values of $52.8 \pm 4.9$ nm and $48.8 \pm 5.6$ nm, respectively (Fig. 3c). We attributed the high and low states to membrane positions associated with the E-cad S-dimer (or monomer) and X-dimer, respectively. This is because the brief duration of the low state in the intensity-time trace corresponds to the short-lived and unstable structure of the X-dimer, and the X-dimer configuration should be observed at a lower height than the S-dimer (Fig. 1a). The low state should not be observed for the *cis*-clustered state as it is a stable state: it has been reported that the rate constant of unbinding *cis*X and *cis*S to X-dimer and S-dimer is around $0.1\ s^{-1}$ (corresponding to a mean unbinding time of 100 s)[16,20]. Furthermore, the "dip" signals in the intensity trace do not result from osmotic or thermally induced membrane fluctuation because these two effects would only lead to a symmetric height distribution (Fig. 3b).

Next, we conducted experiments to validate that the consecutive "dip signals" emanate from the continuous formation of X-dimers within the confocal volume, rather than from the lateral diffusion of X-dimers. This confirms that X-dimers formed within the confocal volume and did not undergo significant diffusion in and out of the confocal volume (Supplementary Fig. 4 and Supplementary Note 5).

It's essential to highlight that during a point measurement with our confocal microscope, the observed signal represents a spatial average over the size of the excitation focus. Since numerous Ecad molecules are present within the confocal spot and not all of them are involved in the conversion process, the measured height values of the membrane cannot be directly converted to the EC5-EC5 distances for a specific dimer state. However, MIET is capable of capturing any small height changes due to its high spatial resolution.

It is essential to note that E-cad-mediated intermembrane adhesion is a multifaceted process with various pathways, including monomer-involved routes. These pathways encompass interactions such as monomer binding to S-/X-dimers, backward unbinding from S-/X-dimers to monomers, and *cis* binding/unbinding of monomers. Unfortunately, in our measurement, we cannot distinguish these specific monomer-involved processes. This limitation arises from the fact that our measurement relies on monitoring membrane height, and monomers exist in a free state without a distinct height signature. Consequently, changes in monomer populations do not induce discernible alterations in membrane height.

For instance, consider the scenario of backward unbinding from an X-dimer to monomers. Given that a significant majority of E-cads are in an S-dimeric state, even if an X-dimer dissociates into monomers within each membrane, the membrane's height would still primarily reflect that of an

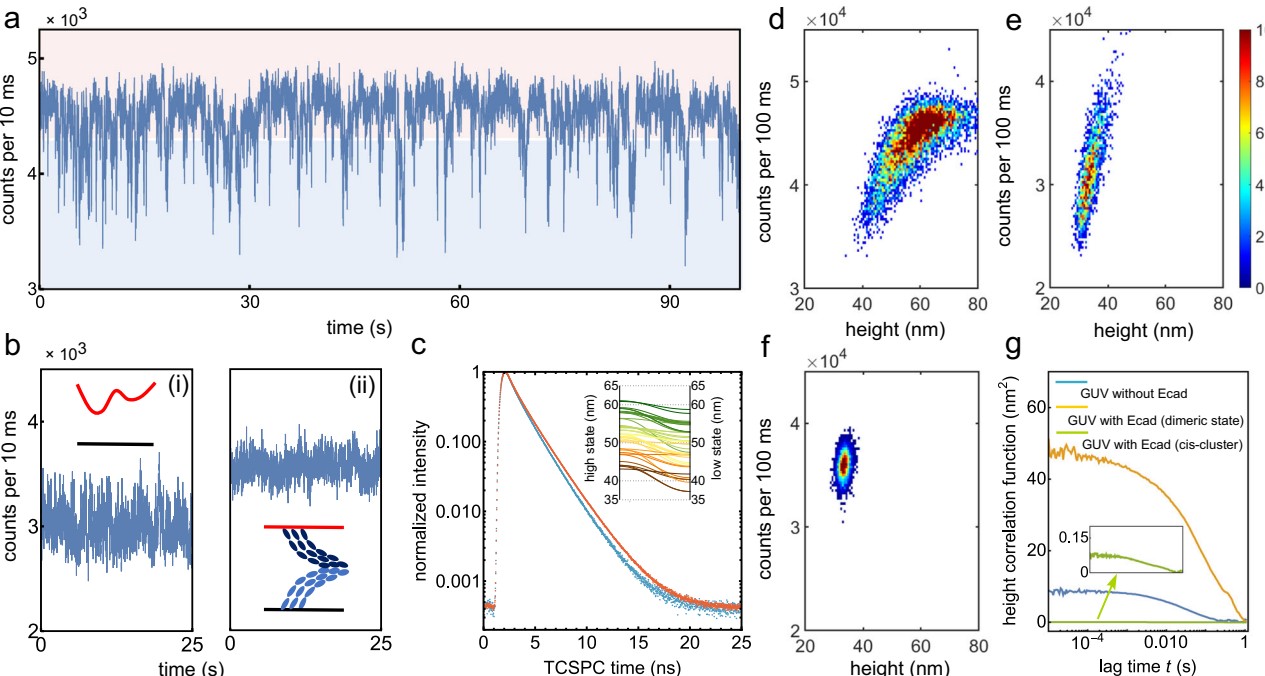

**Fig. 3 | Intensity time trace analysis. a** A typical fluorescence intensity trace recorded from a proximal membrane of highly labeled GUV with E-cad on an E-cad-modified SLB. **b** Fluorescence intensity traces recorded from the GUV-SLB system without any protein (i), and from membranes with *cis*-clusters (ii). **c** Fluorescence lifetime decay curves for the high (red curve) and low (blue curve) states, derived from the trace in (**a**). Inset: calculated heights of high and low states for 45 fluorescence time traces from 27 GUVs. Solid lines represent the height change for

individual fluorescence intensity traces. Two-dimensional histograms of the fluorescence intensity and the height from the surface for (**d**) a membrane with E-cads, **e** a membrane without any proteins, **f** membranes with final *cis*-clusters. **g** Height autocorrelation functions for a membrane without E-cad (blue), a membrane with E-cad but no *cis*-clusters (yellow curve), and a membrane with *cis*-clusters (green curve). Inset shows a zoom of the green curves.

S-dimer. Therefore, we cannot definitively conclude that the abrupt transition from the low state to the high state in the time trace exclusively represents the process of converting X-dimers to S-dimers. This transition encompasses a combination of two potential pathways: X-dimer to S-dimer conversion and X-dimer to monomer conversion.

We further delved into the membrane fluctuations and displacements induced by the formation of X-dimers. To achieve this, we computed membrane heights for a binning time of 100 ms from the intensity trace and conducted an analysis that considered both fluorescence intensity and membrane height fluctuations (as computed from the fluorescence lifetime) throughout the measurement. As illustrated in Fig. 3d, a two-dimensional (2D) histogram reveals correlations between emission intensity and membrane height for the E-cad-modified membranes. This histogram depicts a positive correlation between fluorescence intensity and membrane height, consistent with the theoretical intensity-height curve presented in Fig. 1c.

The height distribution spans a wide range, varying from 40 nm to 75 nm, with a predominant concentration around ~ 60 nm. In contrast, membranes lacking E-cad exhibit a narrower height range, spanning from 30 nm to 45 nm. Moreover, for membranes in the *cis*-clustered state, the height distribution is even more confined, ranging from 32 nm to 36 nm. Notably, the formation of X-dimers leads to increased membrane fluctuations compared to the other two control membranes in the experiment (Fig. 3e, f).

This increased fluctuation arises from a combination of two contributing factors. Firstly, strong thermal fluctuations occur as a result of osmolarity differences. Secondly, dynamic structural changes within dimers contribute significantly to this enhanced fluctuation.

To quantitatively assess membrane fluctuation, we calculated the membrane displacement autocorrelation function ($g_h(t)$) based on the fluorescence intensity autocorrelation function ($g_I(t)$). As demonstrated earlier, fluctuations in fluorescence intensity exclusively originate from axial displacements of the membrane. Consequently, we can convert intensity

correlation curve $g_I(t)$ into a height correlation curve $g_h(t)$ by utilizing the relationship between fluorescence intensity and membrane height (see Materials and Methods). This is done by using the relation $g_I(t) = <\delta h(0) \delta h(t)>$ , where $\delta h(t)$ represents the instantaneous membrane height. The height correlation function provides the root mean square displacement $\psi = \sqrt{\langle \delta h^2 \rangle}$ of the membrane, as well as the relaxation time $\tau^*$, defined as the time point at which $g_h(t)$ has declined to half of its maximum value[36].

As illustrated in Fig. 3g, the displacement amplitude $\psi$ for a GUV membrane with E-cad is $6.9 \pm 1.5$ nm, whereas it is $\psi = 2.9 \pm 0.3$ nm for the membrane lacking E-cad. In the latter case, the amplitude of membrane fluctuations is solely determined by the difference in osmolarity. Following the formation of the final *cis*-clusters, membrane fluctuations become negligible and fall below the detection threshold (Fig. 4g). Furthermore, the relaxation time of the membrane fluctuations for GUVs with E-cad ($\tau^* = 0.1$ s) is notably slower than that for GUVs without E-cad ($\tau^* = 0.04$ s). This difference is attributed to the slower structural interconversion between S- and X-dimers in the presence of E-cad.

To dissect the formation dynamics of the E-cad X-dimer, we individually annotated each intensity time trace by the time spent in the high state (h) and low state (l) between consecutive height switching cycles. These durations are denoted as the waiting time for each state ($t_i$, where $i = h$ or $l$, as shown in Fig. 4a). These waiting times represent the durations required to transition to the next state. For calculating the average times, we first extracted all waiting times (both high and low states) from 25 individual intensity time traces, each lasting 600 s, and then calculated the mean waiting time across all waiting times (>13,000 events): $\langle t_h \rangle = 0.41 \pm 1.29$ s and $\langle t_l \rangle = 0.046 \pm 0.084$ s. Here, $\langle \cdot \rangle$ denotes the averaging time, and errors are determined from the standard deviation of 13,160 switching cycles. The observed large error values in waiting times might be attributed to the fact that E-cads on each membrane are not solely monomers. X-dimers formed from the *cis*-dimers or *cis*-oligomers dramatically differ in stability.

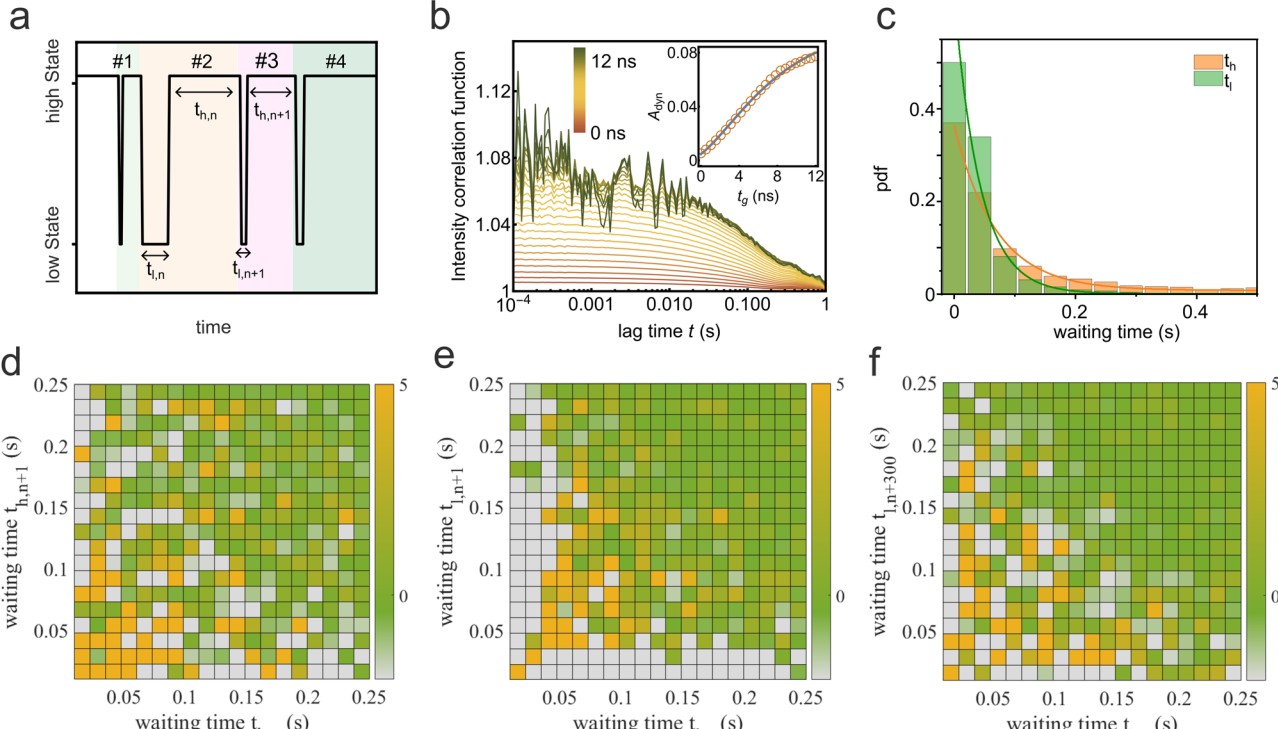

**Fig. 4 | Waiting time correlation analysis. a** We define the waiting time for the low state as $t_l$ and for the high state as $t_h$. These waiting times are calculated as the time intervals between two successive switching cycles. **b** The autocorrelation function is plotted as a function of the window width used for constructing the sg-FCS curves for the fluorescence intensity trace shown in Figure 3a. The sg-FCS correlation functions are color-coded with a gradient from red to green, employing a 0.5 ns step size in increasing window width. These correlation functions were fitted with a mono-exponential decay function, and the resulting correlation amplitudes ($A_{dyn}(t_g)$) are plotted in the panel inset. The data points are fitted with the model of eq. (6). The fitting residuals are shown in Supplementary Fig. 13. **c** Probability distribution functions (pdf) of the waiting times $t_l$ and $t_h$ for more than 10,000 cycles. In (**d, e**), we present 2D difference histograms. These histograms illustrate the correlations between waiting times for consecutive cycles. Specifically, (**d**) shows the 2D difference histogram for $t_h$, and (**e**) shows the same analysis for $t_l$. These plots reveal how waiting times correlate from one cycle to the next. In (**f**), we examine the 2D difference histogram of two waiting times for $t_l$ but with a larger separation of 300 cycles. This analysis explores potential long-time correlations, providing insights into longer-term memory effects.

Additionally, measurements using GUVs functionalized with a higher concentration of E-cad EC show a decrease in the average high state waiting time to $0.25 \pm 0.56$ s, while the low state waiting time does not vary significantly ($0.051 \pm 0.098$ s, Supplementary Fig. 6). This suggests that the measured low state lifetime genuinely reflects the lifetime of the X-dimer.

Using these averaged waiting times, we can calculate the transition rate constants for switching between the low and high states: $k_{l \to h} = 1/\langle t_l \rangle = 22 \pm 39$ s$^{-1}$ and $k_{l \to h} = 1/\langle t_h \rangle = 2.4 \pm 7.6$ s$^{-1}$. This results in an equilibrium constant ($K = k_{h \to l}/k_{l \to h}$) of ~0.11. Intriguingly, the rate constants for converting low state to high state is of the same order of magnitude as the rate constants for conformational switching between the S-dimer and X-dimer in solution, as measured by NMR relaxation dispersion spectroscopy ($k_{X \to S} = 86$ s$^{-1}$)[12]. This further suggests that the transitions between the high and low states in the fluorescence time trace primarily arise from conformational changes between dimers.

As an alternative method for determining the equilibrium constant for the transitions between high and low states, we employed a recently developed method called shrinking gate fluorescence correlation spectroscopy (sg-FCS)[37]. In sg-FCS, one generates a set of fluorescence intensity autocorrelation functions (iACFs) using distinct subsets of photons falling into different time windows (on the nanoseconds time scale) following laser pulses. If dynamic changes are occurring, the amplitudes of the iACFs should incrementally increase with a longer time window.

As depicted in Fig. 4b, the iACFs for the membrane with E-cad clearly exhibit an increase in amplitudes as the time window becomes smaller (in step sizes of 0.5 ns). By fitting the curve amplitudes against window width, we determined the equilibrium constant to be $0.10 \pm 0.013$ (Fig. 4b, inset,

and see section "Methods" for details). Interestingly, this equilibrium constant obtained using sg-FCS closely matches that calculated from the waiting times. This agreement strengthens the conclusion that the dynamic transitions between high and low states primarily result from conformational changes between dimers.

The distribution of individual waiting times follows an exponential decay function for both $t_h$ and $t_l$, both displaying broad distributions (Fig. 4c). The typical trajectory in Supplementary Fig. 9 illustrates the stochastic nature of waiting times for each state. This raises the question of whether these fluctuations in waiting times for each state exhibit temporal correlations. In other words, can the appearance of two consecutive waiting times be statistically correlated with temporal changes in the structures of the dimers?

To investigate these potential correlations, we evaluated the 2D difference histogram, defined as $\delta(t_{i,n}, t_{i,n+1}) = p(t_{i,n}, t_{i,n+1}) - p(t_{i,n}) \times p(t_{i,n+1})$, for the waiting times of each state from two adjacent cycles (Supplementary Note 7). This analysis involves calculating the joint probability for two waiting times, denoted as $p(t_{i,n}, t_{i,n+1})$ (Supplementary Fig. 10), and comparing it to the probability of no correlation between these two times, which is represented by the product of the individual waiting time probabilities, $p(t_{i,n}) \times p(t_{i,n+1})$. This statistical test has previously been employed in the analysis of dynamic disorder in enzymatic catalysis[38–40].

If there is no dynamic disorder, the joint probability $p(t_{i,n}, t_{i,n+1})$ and the product of individual probabilities $p(t_{i,n}) \times p(t_{i,n+1})$ would be the same (Supplementary Fig. 11). If they are different, it hints at the presence of a memory effect induced by conformational changes, where the system 'remembers' its conformation for an extended period.

We conducted an analysis of waiting times from all analyzed trajectories ($N = 25$, each spanning a 600 s experiment). Notably, we observed distinct waiting time distributions for the high and low states.

As depicted in Fig. 4c, d, the waiting times for the high state did not exhibit significant time-correlated features. In contrast, for the low state, we identified a weak and broad diagonal feature in the waiting time distributions. This diagonal feature suggests that longer waiting times are less likely to be followed by shorter ones and vice versa. This intriguing pattern implies a form of temporal memory within the system.

Remarkably, this correlation in waiting times for the low state persisted even when considering cycle lags of 20, corresponding to a time range on the order of seconds (Supplementary Fig. 11). Such memory effects have been previously observed in enzymatic catalysis and attributed to conformational fluctuations in individual enzymes[38–40].

However, in our experimental context, the low states result from the formation of numerous X-dimers, with E-cad molecular densities estimated to be approximately 100–700 molecules per square micrometer within each membrane[23]. Consequently, the observed correlation or memory effect in the low state is less likely to stem from the conformational dynamics of individual X-dimers. Instead, it may arise from inter-dimer communication among the E-cad dimers. Specifically, the presence of X-dimeric states can influence the structure of nearby S-dimers by altering the membrane-membrane distance, potentially affecting the stability and persistence of X-dimers formed from these S-dimers. To further explore the role of membrane fluctuations in E-cad adhesion, we increased the osmolarity difference between the interior of the GUVs and the external buffer from 70 mOsm/L to 170 mOsm/L (Supplementary Fig. 14). This increase led to a corresponding rise in membrane fluctuation amplitude, from $2.9 \pm 0.3$ nm to $6.7 \pm 2.2$ nm. Notably, under these enhanced membrane fluctuations, we observed no transitions between high and low states, and Ecads rapidly formed the final cis-cluster state. This behavior may be attributed to the larger membrane fluctuations weakening the stability of both X- and S-dimers, thereby facilitating the formation of the stable cis-cluster state. The observed dynamics of X-dimer formation and associated membrane displacement may represent early stages of cell-cell adhesion, potentially playing a role in cellular sensing or recognition processes.

## Discussion

We employed MIET spectroscopy to probe membrane-membrane adhesion facilitated by E-cads on each membrane. Our investigations centered on a biomimetic system designed to emulate E-cad-mediated cell adhesion. Specifically, we explored adhesion events of the EC domains of E-cads, which were positioned between SLBs and GUVs.

Consistent with earlier reports[23] highlighting the all-or-nothing nature of E-cad junction formation, our observations revealed two markedly distinct adhesion conformations.

In the first scenario, we observed conventional E-cad-mediated adhesion, resulting in the formation of a cis-cluster between the two membranes. Notably, we observed variations in membrane height during adhesion. Leveraging MIET, we determined membrane height values at different stages of this process. These height values (or membrane-membrane distances), coupled with crystal structure information for E-cad and E-cad junctions, led us to conclude that E-cad experiences a transition from an initial dimeric state with a larger membrane-membrane distance to a final, stable cis-clustered state with a smaller distance.

In contrast, the second scenario involved the observation of only the dimeric state without further clustering. we detected continuous "dip" signals in the fluorescence intensity traces. Through analysis of the EC5-EC5 distance derived from these 'dip' signals, our data strongly support the hypothesis of an X-dimer-dependent mechanism for wild-type E-cad mediated membrane adhesion, with the 'dips' indicative of X-dimer formation. Furthermore, we uncovered the kinetics of X-dimer formation and dissociation. The X-dimer exhibited a brief lifetime of approximately 50 ms, underscoring its role as a short-lived intermediate state.

Our MIET-based analysis extended to quantifying membrane fluctuation amplitudes. We discovered that the conformational dynamics of dimers lead to more pronounced membrane displacements. Interestingly, these membrane displacements triggered a 'memory effect' among X-dimer formations, hinting at mutual cooperativity between membrane displacement and trans-dimers. While prior simulations and experiments have underscored mutual *cis/trans* cooperativity in E-cad, stemming from apparent allosteric enhancement, our results suggest a mechanism whereby membrane displacement also contributes to this cooperativity.

Furthermore, our method presents a approach to studying the stability and dynamics of X-dimers and S-dimers. In our biomimetic system, osmolarity difference and $Ca^{2+}$ concentration emerge as the two critical factors influencing E-cadherin adhesion dimers. For instance, under a larger osmolarity difference (170 mOsm/L), no GUVs exhibited high-state-low-state dynamics, and all GUVs rapidly transitioned to the final cis-cluster state (Supplementary Fig. 14). Moreover, in the absence of $Ca^{2+}$, no E-cadherin-mediated adhesion between GUV and SLB was observed (Supplementary Fig. 15). A follow-up study, employing our MIET method with systematic variations in osmolarity difference and $Ca^{2+}$ concentration, could provide deeper insights into their effects on the dynamics and stability of X-dimers and S-dimers.

In summary, our utilization of MIET spectroscopy and imaging enabled us to monitor the height variations in a model E-cad-modified membrane system. We investigated the kinetic and dynamic processes underlying E-cad-mediated membrane adhesion, pinpointing various E-cad states based on membrane height, and studying their evolution during adhesion. Notably, we unraveled the existence of an X-dimeric intermediate state characterized by its transient nature and its significant impact on membrane displacement, which, in turn, influences the formation and stability of subsequent X-dimers. In particular, we found evidence that X-dimers communicate with one another through membrane displacement. These insights are crucial for understanding the role of E-cad junctions in maintaining cell-cell adhesion, tissue organization, and physiological processes. Further, our findings underscore the potential of MIET spectroscopy and imaging as a potent technique for elucidating subtle structural changes in membrane and membrane protein organization.

## Methods

Details of the materials (lipids and proteins) are available in Supplementary Note 1.

### Ethical Statement

The research conducted in this study complies with all relevant ethical regulations.

### Substrate preparation

The gold-coated coverslips were prepared with the atom vapor deposition method. Briefly, commercial coverslips ($d = 150$ μm) were cleaned by the following detergent treatment: (1) soaking in the piranha lotion for 30 min, (2) ultrasonication in 5% KOH for 15 min, (3) ultrasonication in ethanol for 15 min, and (4) ultrasonication in DI water for 15 min. Steps 2–4 were repeated twice. The cleaned coverslips were further used as substrates for vapor deposition of gold and $SiO_2$ spacer. Under high-vacuum conditions, 2 nm titanium, 10 nm gold, 1 nm titanium, and 10 nm $SiO_2$ were evaporated on the surface of the coverslip layer by layer by using an electron beam source. The lowest deposition rate was maintained at 1 Å s$^{-1}$ to ensure maximum smoothness on the surface.

### Supported lipid bilayer (SLB)

SLBs were prepared with LB-LS techniques with a Langmuir-Blodgett and Langmuir-Schaeffertrough (Nima, Coventry, UK). The subphase was ultrapure water. The proximal monolayer consisted of pure SOPC or DPPC and the distal layer was formed by SOPC or DPPC with 2 mol% PEG2000-DOPE, and 5 mol% DOGS-NTA. For DPPC SLB, 1% NBD-PC was added to the distal layer. The transfer pressure was maintained at 28 mM/m for

SOPC and 30 mM/m for DPPC. The SLBs were kept wet all the time and used directly after preparation. To assess the height/thickness homogeneity of the SLB, we performed graphene-induced energy transfer (GIET) imaging to measure and image the height of the lipid bilayers[31,33,41,42]. GIET, with subnanometer spatial resolution, was chosen over MIET for its higher precision. The resulting height images (h in Supplementary Fig. 12) demonstrate uniform lipid bilayer height (or thickness).

### Giant unilamellar vesicles (GUV)
GUV was prepared by the electro-swelling method. Lipids chloroform mixture consisting of SOPC with 2 mol% PEG2000-DOPE, 1 mol% Atto655-DPPE, and 5 mol% DOGS-NTA was deposited on a home-build electrode and evaporated for 3 h under vacuum at 30 °C. The electrode was assembled into a chamber and filled with 500 μL of 230 mM sucrose solution. Then an alternation current at 15 Hz for 3 h and a peak-to-peak voltage of 1.6 followed by 8 Hz for 30 min were applied to the chamber. The GUVs suspension was stored at 4 °C and used within 3 days.

### Measurement sample preparation
For preparing the measurement sample, SLB was first incubated with 2 μM $NiSO_4$ PBS solution for 15 min and then washed with PBS 5 times. Afterward, this SLB was further exposed to 10 μg/mL E-cad PBS solution for 3 h. Meanwhile, the stock GUVs solution was also incubated with the same concentrations of $NiSO_4$ and E-cad for 3 h. Then the E-cad-functionalized GUVs were diluted 100 times with PBS-$CaCl_2$ buffer (140 mM NaCl, 3 mM KCl, 10 mM $Na_2HPO_4$, 2 mM $KH_2PO_4$, 750 μM $CaCl_2$, pH 7.4). Finally, the diluted GUVs solution was added into the chamber with SLBs, and measurements were started immediately. To avoid solution evaporation, a glass slide was used to cover the chamber during the measurement.

### MIET measurement
All measurements were carried out with home-built confocal microscopy equipped with a time-correlated single photon counting (TCSPC) device. A pulsed diode laser ($\lambda_{exc}$ = 640 nm, LDH-D-C 640, PicoQuant), which has a pulse width of 50 ps FWHM and repetition rate of 40 MHz, and a clean-up filter (LD01-640/8 Semrock) worked as the excitation source. A beam of 12 mm diameter, which was collimated from an infinity-corrected 4× objective (UPISApo 4X, Olympus), was reflected by a dichroic mirror (Di01-R405/488/561/635, Semrock) towards a high numerical aperture objective (UApoN 100X, oil, 1.49 N.A., Olympus). The emission light was focused into a pinhole of 100 μm diameter and refocused onto two avalanche photodiodes ($\tau$-SPAD, PicoQuant). Additionally, a long-pass filter (BLP01-647R-25, Semrock) and two band-pass filters (Brightline HC692/40, Semrock) were used before the pinhole and the detectors, separately. Photo signals acquired from the detector were processed by a multi-channel picosecond event timer (Hydraharp 400, PicoQuant) with 16 ps time resolution. In a typical measurement, we scanned the sample at the coverslip surface to determine the contour of the GUV and choose the appropriate point position for data acquisition. The power of the laser is adjusted until a maximum count rate of 100–500 kcps is reached.

### Principle of MIET
The theory for converting the lifetime values to the substrate-fluorophore distances with MIET has been elaborated in detail in several previous publications[30–32]. Briefly, the electromagnetic field of a fluorescing molecule (electric dipole emitter) is represented by a superposition of plane electromagnetic waves. When interacting with the MIET substrate, each plane wave is reflected and transmitted based on Fresnel's reflection and transmission laws. The full field is thus represented by a superposition of the incident and reflected (molecule's half space) or transmitted plane waves (objective's half space). The emission rate of the fluorescent molecule is then calculated by integrating the Poynting vector of the full field over a closed surface enclosing the emitter. Thus, we get an expression for the radiative

emission power ($S(\theta, z_0)$) which can be decomposed as

$$S(\theta, z_0) = S_\perp \cos^2\theta + S_\parallel \sin^2\theta \qquad (1)$$

where $z_0$ is the distance of the emitter from the surface, $\theta$ is the orientation angle of this emitter (angle between emission dipole axis and vertical axis), and $S_\perp$ (or $S_\parallel$) are the radiative emission rates of a dipole emitter oriented perpendicular or parallel to the substrate, respectively. Taking the molecule's nonradiative transition into account, the expression for the fluorescence lifetime ($\tau_f(\theta, z_0)$) of a molecule is given by

$$\frac{\tau_f(\theta, z_0)}{\tau_0} = \frac{S_0}{\varphi S(\theta, z_0) + (1 - \varphi)S_0} \qquad (2)$$

where $\tau_0$ is the lifetime of the molecule in free space without any MIET substrate, $\varphi$ is the quantum yield of the molecule, and $S_0$ is the emission power the emitter in free space. Furthermore, the relative brightness $b(\theta, z_0)$ of an emitter near the metal surface is proportional to the product of the position- and orientation-dependent collection efficiency $\eta(\theta, z_0)$ of fluorescence detection and the local quantum yield $\varphi_{local}(\theta, z_0)$. The former is found by integrating the position- and orientation-dependent angular distribution of emission of the emitter over the cone of light collection of the microscope's objective. The local quantum yield $\varphi_{local}(\theta, z_0)$ is defined by

$$\varphi_{local}(\theta, z_0) = \frac{\varphi S(\theta, z_0)}{\varphi S(\theta, z_0) + (1 - \varphi)S_0} \qquad (3)$$

Thus, we find

$$b(\theta, z_0) = \eta(\theta, z_0)\varphi_{local}(\theta, z_0) \qquad (4)$$

Figure 2c shows the calculated lifetime and relative brightness as a function of distance $z$ for the case of an emitter oriented randomly to the substrate. For the DPPE-Atto655 labeled membrane, the free space lifetime was measured on a GUV sample on a glass surface without gold film, and the quantum yield ($\varphi = 0.36$) was measured using a nanocavity method[43]. For the fluctuating membrane, we used random orientation to calculate the calibration curves of lifetime and relative brightness because of the fluctuations-induced deformations in lipid bilayers[33,44]. Note that when the membrane forms the final *cis*-cluster state, the membrane does not fluctuate much. In this case, we assumed a dye orientation parallel to the substrate surface when calculating the calibration curve[31].

### Conversion of intensity to height correlation curves
In general, the autocorrelation function for conventional FCS is defined as

$$g_I(t) = \frac{\langle \delta I(t')\delta I(t' + t)\rangle_{t'}}{\langle I \rangle^2} \qquad (5)$$

while the height correlation function ($g_h$) for the membrane fluctuations is defined as $g_h(t) = \langle \delta h(t')\delta h(t' + t)\rangle_{t'}$, where $\delta h(t)$ is the membrane displacement from its average position.

The relationship between fluorescence intensity ($I$) and membrane height ($h$) can be determined from eq.(4). However, to simplify the conversion, the $I$–$h$ relationship can be estimated approximately from the experiment. Supplementary Fig. 7 shows a two-dimensional histogram of the fluorescence intensity and fitted lifetimes measured from the sample without E-cad between GUV and SLB, it can be found that the dependence of intensity on height is almost linear. Thus we take a linear fitting on the counts versus heights data, which gives the slope ($m = \delta I/\delta h$). Finally, we obtained the $g_h(t)$ from the conversion of $g_I(t)$:

$$g_h(t) = \langle \delta h(0)\delta h(t)\rangle = \frac{\langle I(t)\rangle^2}{m^2}g_I(t) \qquad (6)$$

## sg-FCS analysis

The theory for analyzing the sg-FCS has been thoroughly elaborated by Schröder et al.[37]. In brief, each photon is recorded with two-time tags, namely the macro-time and micro-time, which respectively refer to the photon's arrival time from the beginning of the experiment and the delay time between the laser pulse and the photon detection. We generated a series of sg-FCS curves (iACFs) using different subsets of photons depending on their micro-time. Then, the bunching amplitude ($A_{dyn}$) for each iACF was obtained by fitting a mono-exponential model. At last, we determined the equilibrium constant ($K$) by fitting the correlation amplitude versus sg-FCS time window width using the following model equation

$$A_{dyn}(t_g) = K \left[ \frac{1 - e^{-b(t_g - t_0)}}{1 + K \cdot e^{-b(t_g - t_0)}} \right]^2 \qquad (7)$$

where $t_0$ is the temporal position of the laser excitation pulse peak, and $b$ denotes an energy transfer rate constant. It's important to note that all iACFs were background-corrected by measuring the background signal on an empty gold-coated coverslip.

## Reporting summary

Further information on research design is available in the Nature Portfolio Reporting Summary linked to this article.

## Statistics and reproducibility

All values are expressed as the mean ±SD. No statistical methods was used to predetermine sample. No data were excluded from the analysis. All experiments were repeated at least twice with reproducible results.

## Data availability

There are no restrictions on data availability. All data of the figures (Figs. 1, 2, 3, and 4) in this manuscript together with Matlab and Mathematica programs that were used to generate these figures are publicly available in the figshare repository: https://doi.org/10.6084/m9.figshare.27646728[45]. Unprocessed raw data (.ptu file) are also included in this figshare repository. Upon request, all raw data files can be requested from the first author Tao Chen (tao.chen@phys.uni-goettingen.de). Requests will be fulfilled within 4 weeks.

## Code availability

All codes used for analyzing the raw data are deposited on GitHub at https://gitlab.gwdg.de/tchen1/ecadherindynamiet. In particular, the depository contains: • *Matlab* (v. 2022b, MathWorks® Inc.) code used for calculating the calibration curves (Fig. 1c, d), extracting the data from raw .ptu file and calculating the intensity time traces (or height trace) for Figs. 2d, e, 3a, b, and conducting the lifetime calculation, sg-FCS analysis (Fig. 4b), correlation analysis (Fig. 3g), intensity-height 2D histogram (Fig. 3d–f) and waiting time analysis (Fig. 4d–f); • A *Mathematica* (v. 13.2.1.0 Wolfram Research Inc.) notebook that generates the graphs of Figs. 1, 2d, e, 3a–c, g, and 4b, c; • A Inventory file and a LifetimeL file in the depository to explain the codes. The FLIM images (Figure 2a) are analyzed by using a published software (*TrackNTrace* Lifetime Edition)[46]. Additionally, all codes are citable from Zenodo (DOI: 10.5281/zenodo.14066650)[47].

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

## Acknowledgements

J.E. acknowledges financial support by the DFG through Germany's Excellence Strategy EXC 2067/1-390729940. T.C. and J.E. thank the European Research Council (ERC) for financial support via project "smMIET" (grant agreement no. 884488) under the European Union's Horizon 2020 research and innovation program.

## Author contributions

T.C. prepared all samples, and performed all the measurements. T.C. and N.K. analyzed the data. All authors helped with preparing the manuscript, which was finalized by J.E.

## Funding

## Competing interests

The authors declare no competing interests.
