## [Transparent Peer Review file · Communications Biology]

Observation of E-cadherin Adherens Junction Dynamics with Metal-Induced Energy Transfer Imaging and Spectroscopy

Corresponding Author: Dr Tao Chen

Version 0:

Reviewer comments:

Reviewer #1

(Remarks to the Author)

In the article "Observation of E-cadherin Adherens Junction Dynamics with Metal-Induced Energy Transfer Imaging and Spectroscopy" Chen et al., use a technique called MIET spectroscopy to study how E-cadherin (E-cad) proteins on cell membranes interact with each other. The study reveals two different adhesion scenarios. In the first, E-cads formed a stable, clustered connection between the membranes. In the second scenario, the E-cads only formed a temporary connection. By analysing the distance between the membranes using metal induced energy transfer (MIET), the authors were able to measure the distance between the membranes at different stages of this process and concluded that E-cad goes from a temporary extended state to a more permanent compact state. They also found evidence for a short-lived intermediate state called an X-dimer and observed that the formation of X-dimers caused the membranes to move, and this movement seemed to influence the formation of other X-dimers. These findings provide new insights into the dynamics of E-cad mediated adhesion and the potential of this technique (MIET) for studying such interactions.

Overall, the paper is technically solid, uses a new technique to measure the dynamics of an important biological system, and offers new insights into E-cad mediated adhesion. I highly recommend publication of this article after the authors address the following points:

1. Are the lipid bilayers of uniform thickness?
2. When measuring the brightness data in Fig. 1c has there been a correction performed for photobleaching. It is possible some molecules will bleach over time. Can the authors comment on this point.
3. Throughout the manuscript (except once) when noting the estimated heights from the FLIM images the authors do not provide any error ranges. It would be useful to have an error range to meaningfully interpret the height data.
4. The authors write "by averaging the waiting times across all intensity time traces". How was the averaging done? If the average times are estimated from exponential fits to the dwell time histograms, it would be good to check whether a 2-exponential function better fits the data.
5. Did the authors think of using something like a hidden Markov Model (HMM) for fitting the time traces to check if additional states other than the assigned "high" or "low" states exist. It looks from the trace in Fig. 3a there are more than just two-states.
6. Error ranges need to be reported for the transition rates.

Reviewer #2

(Remarks to the Author)

This is an interesting manuscript, which provides an in-depth analysis of E-cadherin (E-cad)-mediated cell adhesion by utilizing Metal-Induced Energy Transfer (MIET) imaging and spectroscopy. The authors have introduced an innovative methodology to examine changes in the distance between cell membranes throughout the adhesion process. Their focus is on understanding the formation and stability of various E-cadherin states, including monomers, X-dimers, and S-dimers. This research offers significant insights into the molecular mechanisms underlying adherens junction formation and highlights the impact of membrane dynamics on these processes. The entire manuscript is sufficiently documented, easy to follow, and well explained. However, some issues should be addressed before the manuscript is suitable for publication.

1. The paper indicates that membrane fluctuations play a role in E-cad adhesion. Can the authors explore this further by quantifying the impact of these fluctuations on the stability of different E-cad states (e.g., monomers, X-dimers, S-dimers), and how they influence the overall adhesion process?

2. For the benefit of the readership of Communications Biology, the authors should offer some comments on how well the biomimetic system used in the study replicates the in vivo environment of E-cadherin-mediated adhesion. Also, could the observed distance variations be linked to specific cellular or molecular events during adhesion, such as signal transduction or junction formation?
3. There should be more emphasis on the statistics and fitting methods description. Some traces in figure 3 seem to suggest more than two states. The authors should provide some basis of this model (like giving residual distributions etc) and show that by adding another component does not improve the fits. There should be some error estimations in the legends of the figures?
5. How might the findings regarding X-dimer dynamics and membrane displacement be relevant to understanding pathological conditions, such as cancer metastasis or epithelial dysfunction? Are there specific disease models that could benefit from these insights?
6. This may be useful to know if varying experimental conditions like, ionic strength, pH etc would influence the fluctuations and dynamics of X-dimers and S-dimers? Are there specific conditions under which these effects are amplified or reduced?

Version 1:

Reviewer comments:

Reviewer #1

(Remarks to the Author)

I have carefully read the rebuttals provided by the authors. They have now answered all the questions in a comprehensive manner. I recommend publication of this manuscript in its current form.

Reviewer #2

(Remarks to the Author)

The authors have addressed all my concerns. I have no other comments.

Dr. Tao Chen, Prof. Jörg Enderlein
Drittes Physikalisches Institut – Biophysik
Friedrich-Hund-Platz 1· 37077 Göttingen

Tel.: +49 (0) 551 – 39 26908
Secretary: +49 (0) 551 – 39 27714
Fax: +49 (0) 551 – 39 27720
Cell phone: +49 (0) 178 – 538 1130
Email: jenderl@gwdg.de;
Tao.chen@phys.uni-goettingen.de
<https://www.joerg-enderlein.de>

Response to the reviewer's comment to our manuscript "Observation of E-cadherin Adherens Junction Dynamics with Metal-Induced Energy Transfer Imaging and Spectroscopy" by Tao Chen, Narain Karedla, and Jörg Enderlein

Reviewers' comments:

Reviewer #1 (Remarks to the Author):

In the article "Observation of E-cadherin Adherens Junction Dynamics with Metal-Induced Energy Transfer Imaging and Spectroscopy" Chen et al., use a technique called MIET spectroscopy to study how E-cadherin (E-cad) proteins on cell membranes interact with each other. The study reveals two different adhesion scenarios. In the first, E-cads formed a stable, clustered connection between the membranes. In the second scenario, the E-cads only formed a temporary connection. By analysing the distance between the membranes using metal induced energy transfer (MIET), the authors were able to measure the distance between the membranes at different stages of this process and concluded that E-cad goes from a temporary extended state to a more permanent compact state. They also found evidence for a short-lived intermediate state called an X-dimer and observed that the formation of X-dimers caused the membranes to move, and this movement seemed to influence the formation of other X-dimers. These findings provide new insights into the dynamics of E-cad mediated adhesion and the potential of this technique (MIET) for studying such interactions.

Overall, the paper is technically solid, uses a new technique to measure the dynamics of an important biological system, and offers new insights into E-cad mediated adhesion. I highly recommend publication of this article after the authors address the following points:

We thank the reviewer for her/his comments that helped us to improve our manuscript.

1. Are the lipid bilayers of uniform thickness?

Yes, the thickness of the lipid bilayers is very uniform. We prepared the lipid bilayers (SOPC and DPPC/NBD-PC) with E-cadherins using the Langmuir-Blodgett (LB) and the Langmuir-Schaeffer (LS) method, and incorporated DPPE-atto655 into the bilayers and determined the axial positions of the dyes (see Fig. R1) with graphene-induced energy transfer (GIET) imaging (Ghosh, A., Nat Protoc 16, 3695–3715 (2021)). The proximal leaflet, composed of pure lipids (SOPC or DPPC/NBD-PC) and DPPE-atto655, was prepared via the LB method. The distal leaflet, containing SOPC or DPPC/NBD-PC, DPPE-atto655, DOPE-PEG2000, and DGS-NTA, was prepared using the LS method. From the obtained dye positions, we calculated the mean height of the lipid bilayers (average of bottom and top leaflet). The resulting height images (h in Fig. R1) show a very uniform lipid bilayer height. It should be noted that areas with reduced height values in Fig. R1 are attributed to defects or regions containing more than a single graphene monolayer (please see SEM image from the manufacturer: <https://eu.graphenea.com/products/easy-transfer-monolayer-graphene-on-polymer-film-1-cm-x-1-cm>), and not to heterogeneity in the bilayer itself.

We added a corresponding paragraph to the Methods section in the main text, and added Supplementary Figure 12 showing GIET measurement.

Fig.R1. Graphene-induced energy transfer (GIET) imaging for determining average SLB heights. (a) A schematic illustration of the GIET measurement setup is shown. SLBs composed of SOPC or DPPC/NBD-PC were deposited on a graphene monolayer with a 5-nm silica spacer. To measure membrane height, a small fraction of lipids was labeled at the headgroup with the fluorophore Atto655. GIET enables precise determination of the mean SLB height (h), corresponding to the average axial position of the dyes in the bottom and top leaflets, with sub-nanometer resolution. (b) Height maps are provided for both SOPC-SLB (left) and DPPC/NBD-PC-SLB (right), revealing a nearly uniform height distribution across the SLBs. Areas with reduced height are attributed to defects or regions with multiple graphene layers.

2. When measuring the brightness data in Fig. 1c has there been a correction performed for photobleaching. It is possible some molecules will bleach over time. Can the authors comment on this point.

We did not do any photobleaching correction. In our measurements, the utilization of high labeling concentrations (1 mol%) allows us to use much lower excitation power as compared to conventional methods (at least 100 times lower), effectively eliminating photobleaching in all our measurements. More long-time traces can be found in Figure S6 in SI, no intensity decays were observed.

3. Throughout the manuscript (except once) when noting the estimated heights from the FLIM images the authors do not provide any error ranges. It would be useful to have an error range to meaningfully interpret the height data.

We added error values to the main text (standard errors of the mean).

4. The authors write “by averaging the waiting times across all intensity time traces”. How was the averaging done? If the average times are estimated from exponential fits to the dwell time histograms, it would be good to check whether a 2-exponential function better fits the data.

For calculating the average times, we first extracted all waiting times (high or low) from 25 individual intensity time traces, each lasting 600 s, and then calculated the mean waiting time for each state (> 13,000 events). We have modified the sentence in the main text to clarify the averaging method.

5. Did the authors think of using something like a hidden Markov Model (HMM) for fitting the time traces to check if additional states other than the assigned “high” or “low” states exist. It looks from the trace in Fig. 3a there are more than just two-states.

We did attempt to use a Hidden Markov Model (HMM) to fit the time traces, as shown in Fig. R2. However, even when using a four-state model, the resulting fit was unsatisfactory. This issue arises because the time traces reflect membrane position but not E-cadherin (Ecad) positions themselves. As a result, two additional factors contribute to the intensity fluctuations: (1) vertical fluctuations of the membrane, which lead to a broad intensity distribution; and (2) spatial averaging over the size of the excitation focus, where more than one X-dimer form can contribute to the signal.

Therefore, we assigned only two states to the time trace, a “high” and a “low” state. Indeed, it is possible that the high state has contributions from two different E-cad states, the monomer and the S-dimer state. We tried to explain the complexity of the system and measurement in the original version of manuscript as follows: ‘It is essential to note that E-cad-mediated intermembrane adhesion is a multifaceted process with various pathways, including monomer-involved routes. These pathways encompass interactions such as monomer binding to S-/X-dimers, backward unbinding from S-/X-dimers to monomers, and cis binding/unbinding of monomers. Unfortunately, in our measurement, we cannot distinguish these specific monomer-involved processes. This limitation arises from the fact that our measurement relies on monitoring membrane height, and monomers exist in a free state without a distinct height signature. Consequently, changes in monomer populations do not induce discernible alterations in membrane height.

For instance, consider the scenario of backward unbinding from an X-dimer to monomers. Given that a significant majority of E-cads are in an S-dimeric state, even if an X-dimer dissociates into monomers within each membrane, the membrane's height would still primarily reflect that of an S-dimer. Therefore, we cannot definitively conclude that the abrupt transition from the low state to the high state in the time trace exclusively represents the process of converting X-dimers to S-dimers. This transition encompasses a combination of two potential pathways: X-dimer to S-dimer conversion and X-dimer to monomer conversion.’

As HMM proved to be unsuitable for our data analysis, we explored alternative methods for analyzing the time traces. Because the EC5-EC5 distance measured during the short-living low state is consistent with that of the X-dimer, we attribute this state to the X-dimer conformation.

Thus, our analysis focused on determining the exact duration of the X-dimer conformation. We employed a double-threshold method to define the start and end times of the low and high states within the intensity-time traces (for details, see Supplementary Figure 8 and Supplementary Note 6). Our analysis indicates that only 2% of the signals in the high state were mistakenly identified as coming from the low state.

Fig R2. Exemplary time trace with corresponding HMM fitting.

6. Error ranges need to be reported for the transition rates.

We added error values for the transition rates to the main text.

Reviewer #2 (Remarks to the Author):

This is an interesting manuscript, which provides an in-depth analysis of E-cadherin (E-cad)-mediated cell adhesion by utilizing Metal-Induced Energy Transfer (MIET) imaging and spectroscopy. The authors have introduced an innovative methodology to examine changes in the distance between cell membranes throughout the adhesion process. Their focus is on understanding the formation and stability of various E-cadherin states, including monomers, X-dimers, and S-dimers. This research offers significant insights into the molecular mechanisms underlying adherens junction formation and highlights the impact of membrane dynamics on these processes. The entire manuscript is sufficiently documented, easy to follow, and well explained. However, some issues should be addressed before the manuscript is suitable for publication.

We thank the reviewer for her/his comments that helped us to improve our manuscript.

1. The paper indicates that membrane fluctuations play a role in E-cad adhesion. Can the authors explore this further by quantifying the impact of these fluctuations on the stability of different E-cad states (e.g., monomers, X-dimers, S-dimers), and how they influence the overall adhesion process?

In our original manuscript, we reported that no adhesion was observed in GUVs when there was no osmolarity difference between the inside and outside of the vesicles (tense GUVs with minimal membrane fluctuation, see Supplementary Figure 3). For the adhesion experiments detailed in our manuscript, we introduced an osmolarity difference of 70 mOsm/L, which enhanced membrane fluctuations, resulting in a fluctuation amplitude of 2.9 nm in membranes without E-cadherin (see Figure 3g in the main text). To further investigate the impact of membrane fluctuations on E-cadherin adhesion, we conducted additional experiments by increasing the osmolarity difference to 170 mOsm/L, thereby generating larger membrane fluctuations with an amplitude of 6.7 nm (Figure R3). Under these conditions, the E-cadherin in all GUVs rapidly formed the final cis-cluster state, and no transitions between high and low intensity states were observed (Figure R3). This suggests that increased membrane flexibility reduces the stability of X- and S-dimers, favoring the formation of stable cis-cluster states. These findings further support our conclusion that membrane fluctuations play a crucial role in E-cadherin-mediated adhesion.

We included additional data in the Supplementary Information and added the following discussion to the main text: 'To further explore the role of membrane fluctuations in E-cad adhesion, we increased the osmolarity difference between the interior of the GUVs and the external buffer from 70 mOsm/L to 170 mOsm/L (Supplementary Figure 14). This increase led to a corresponding rise in membrane fluctuation amplitude, from 2.9 ± 0.3 nm to 6.7 ± 2.2 nm. Notably, under these enhanced membrane fluctuations, we observed no transitions between high and low states, and Ecads rapidly formed the final cis-cluster state. This behavior may be attributed to the larger membrane fluctuations weakening the stability of both X- and S-dimers, thereby facilitating the formation of the stable cis-cluster state.'

Figure R3. (a) Left: Schematic illustration of the control measurement of the fluorescently-labeled GUVs containing E-cad on SLB. The osmolarity difference between the interior of the GUVs (230 mOsm/L, sucrose solution) and the external buffer (400 mOsm/L, phosphate-buffered saline, PBS, mixed with 0.75 mM CaCl₂ solution) is 170 mOsm/L. Right: Corresponding FLIM image captured 10 minutes after incubating the GUVs in the chamber. Multiple areas within the chamber were examined, and no GUVs exhibited the X-dimer dynamic transition (as described in the second scenario in the main text). (b) Left: Schematic illustration of the control measurement of the fluorescently-labeled GUVs without E-cad on SLB. The bottom membrane of GUV undergoes fluctuation. Right: The calculated height correlation function, indicating the membrane fluctuation amplitude of 6.7 ± 2.2 nm.

2. For the benefit of the readership of Communications Biology, the authors should offer some comments on how well the biomimetic system used in the study replicates the in vivo environment of E-cadherin-mediated adhesion. Also, could the observed distance variations be linked to specific cellular or molecular events during adhesion, such as signal transduction or junction formation?

The plasma membrane is highly complex, with its array of lipids, proteins, and carbohydrates contributing to a wide range of interactions and functions. This complexity makes it challenging to isolate and study specific interactions. In contrast, the biomimetic system we used here, comprising model membranes and E-cadherin, offers a simplified and controlled environment that allows us to focus exclusively on the specific mechanism of E-cadherin-mediated adhesion. While this system lacks the complexity of biological membranes and does not fully replicate the intricate behavior seen in cellular contexts, it enables us to observe E-cadherin interactions without the confounding influences of other cellular proteins, contextual factors, or active processes. This reductionist approach provides clear insights into the fundamental aspects of E-cadherin-mediated adhesion, isolated from the broader complexity of live cell systems. We added to the main text the sentence: 'It should be noted that while the biomimetic system lacks the complexity of biological membranes and does not fully replicate the intricate behavior seen in cellular contexts, it enables us to observe E-cad interactions without the confounding influences of other cellular proteins, contextual factors, or active processes.'

Regarding the question 'could the observed distance variations be linked to specific cellular or molecular events during adhesion, such as signal transduction or junction formation?' our answer is:

Yes, variations in intermembrane distance clearly affect junction formation, while, conversely, junction formations also determine the intermembrane distance. Additionally, the dynamics of intermembrane distance play an active role in facilitating junction formation. For instance, as shown in the main text (Fig. 4), we illustrate how memory effects as reflected by waiting time correlations influence X-dimer formation. This memory effect may be due to the following feedback mechanism: when an X-dimer forms, it reduces the intermembrane distance. This, in turn, decreases the stability and persistence of nearby S-dimers, which favor a larger intermembrane distance. As a result, these destabilized S-dimers are more likely to convert into X-dimers, further influencing the junction dynamics. Furthermore, after forming the final cis-cluster state (Fig. 3), the intermembrane distance becomes very small and almost no distance variations are observable, suggesting a strong interplay of membrane-membrane distance and junction formation.

3. There should be more emphasis on the statistics and fitting methods description. Some traces in figure 3 seem to suggest more than two states. The authors should provide some basis of this model (like giving residual distributions etc) and show that by adding another component does not improve the fits. There should be some error estimations in the legends of the figures?

In the main text, we added detailed descriptions of lifetime determination and fitting, as well as sg-FCS fitting. When analyzing the time traces, we did not apply any fitting model, as explained in our response to point #5 of the first reviewer. We have included standard errors for all our statistical analyses and added residual value curves for all fits shown in the Supplementary Information.

5. How might the findings regarding X-dimer dynamics and membrane displacement be relevant to understanding pathological conditions, such as cancer metastasis or epithelial dysfunction? Are there specific disease models that could benefit from these insights?

*The formation of X-dimers lowers the energy barrier between S-dimers and E-cadherin monomers, facilitating transitions between these states. Notably, previous studies have shown that cells with X-dimer mutations exhibit non-adhesive behavior (Yuliya I. Petrova et al., Molecular Biology of the Cell, 2016). We propose that the observed dynamics of X-dimer formation and the accompanying membrane displacement represent early stages of cell-cell adhesion, potentially playing a key role in cellular sensing and recognition. In metastatic cancer, the loss or dysfunction of E-cadherin disrupts cell-cell junctions, leading to decreased adhesion and enabling cancer cells to dissociate and metastasize. The modulation of intermembrane distance through X-dimer formation, as demonstrated in this study, could offer valuable insights into how cancer cells alter membrane dynamics and adhesion to promote metastasis. To strengthen the biological implications of our findings, we added the following to the main text: **'The observed dynamics of X-dimer formation and associated membrane displacement may represent early stages of cell-cell adhesion, potentially playing a role in cellular sensing or recognition processes.'***

Since E-cadherin is widely expressed in epithelial cells and many cancer types, our research may help uncover new mechanisms for targeting cell adhesion and potentially impeding metastasis. However, as a study focused on molecular structural biology, our primary aim is to elucidate molecular mechanisms rather than explore specific biological applications. While these findings provide fresh perspectives on the fundamental biology of cell adhesion, their direct application to disease models will require further research and experimental validation.

6. This may be useful to know if varying experimental conditions like, ionic strength, pH etc would influence the fluctuations and dynamics of X-dimers and S-dimers? Are there specific conditions under which these effects are amplified or reduced?

We agree with the reviewer that it is crucial to investigate the factors influencing the dynamics of X-dimers and S-dimers. In our measurements, we believe the two most important factors are the osmolarity difference and Ca^{2+} concentration. Regarding the osmolarity difference, we have included additional results showing how the osmolarity difference affects membrane fluctuation amplitudes and have discussed its effect on E-cadherin adhesion (please see the response to your first point). As for the effect of Ca^{2+} concentration, we conducted additional measurements using a buffer without Ca^{2+} . As shown in Figure R4, no E-cadherin-mediated adhesion between GUV and SLB was observed in the absence of Ca^{2+} .

We have included this additional data in the Supplementary Information and added more discussion to the main text: *'Furthermore, our method presents a novel approach to studying the stability and dynamics of X-dimers and S-dimers. In our biomimetic system, osmolarity difference and Ca^{2+} concentration emerge as the two most critical factors influencing E-cadherin adhesion dimers. For instance, under a larger osmolarity difference (170 mOsm/L), no GUVs exhibited high-state-low-state dynamics, and all GUVs rapidly transitioned to the final cis-cluster state (Supplementary Figure 14). Moreover, in the absence of Ca^{2+} , no E-cadherin-mediated adhesion between GUV and SLB was observed (Supplementary Figure 15). A follow-up study, employing our MIET method with systematic variations in osmolarity difference and Ca^{2+} concentration, could provide deeper insights into their effects on the dynamics and stability of X-dimers and S-dimers.'*

Fig. R4. Fluorescence images of E-cad-modified GUV on E-cad-modified SLB recorded at two different focal planes: (a) 10 μm above the surface and (b) directly at the surface. The images on the right were recorded 20 seconds after the images on the left. White arrows indicate the direction of vesicle movement. (c) Schematic representation of the E-cad-modified deflated GUV on E-cad-modified SLB, above a gold surface with 10 nm silica spacer. GUVs were prepared by immersing them into a PBS solution without Ca^{2+} (400 mOsm/L).